# Arbitrary Optics for Gaussian Splatting Using Space Warping

**DOI:** 10.3390/jimaging10120330

**Published:** 2024-12-22

**Authors:** Jakob Nazarenus, Simin Kou, Fang-Lue Zhang, Reinhard Koch

**Affiliations:** 1Department of Computer Science, Kiel University, 24118 Kiel, Germany; rk@informatik.uni-kiel.de; 2School of Engineering and Computer Science, Victoria University of Wellington, Wellington 6012, New Zealand; simin.kou@vuw.ac.nz (S.K.); fanglue.zhang@vuw.ac.nz (F.-L.Z.)

**Keywords:** 3D reconstruction, novel view synthesis, 3D Gaussian Splatting, camera models

## Abstract

Due to recent advances in 3D reconstruction from RGB images, it is now possible to create photorealistic representations of real-world scenes that only require minutes to be reconstructed and can be rendered in real time. In particular, 3D Gaussian splatting shows promising results, outperforming preceding reconstruction methods while simultaneously reducing the overall computational requirements. The main success of 3D Gaussian splatting relies on the efficient use of a differentiable rasterizer to render the Gaussian scene representation. One major drawback of this method is its underlying pinhole camera model. In this paper, we propose an extension of the existing method that removes this constraint and enables scene reconstructions using arbitrary camera optics such as highly distorting fisheye lenses. Our method achieves this by applying a differentiable warping function to the Gaussian scene representation. Additionally, we reduce overfitting in outdoor scenes by utilizing a learnable skybox, reducing the presence of floating artifacts within the reconstructed scene. Based on synthetic and real-world image datasets, we show that our method is capable of creating an accurate scene reconstruction from highly distorted images and rendering photorealistic images from such reconstructions.

## 1. Introduction

Based on the foundational work in the field of 3D reconstruction, in recent years, the topic of Novel View Synthesis (NVS) has gained momentum due to the rapid progress of learning-based methods [1]. Using a differentiable scene representation in combination with a differentiable rendering pipeline, these methods have achieved state-of-the-art results in recreating photorealistic scenes from images [2]. As the underlying models were trained to encode view-dependent effects, the models outperform renderings from classical 3D reconstruction models with respect to visual fidelity, enabling the recreation of complex lighting scenarios. Since then, more recent contributions have mitigated one of the most significant downsides of learning-based NVS methods: their training and rendering times [3,4]. One method in particular, 3D Gaussian Splatting (3DGS), established a significant improvement over the current state of the art by introducing an efficient scene representation and rendering pipeline [5]. 3DGS enables real-time rendering and training scenes in several minutes while achieving a very high degree of detail. Its efficiency and visual quality have led to its adoption in VR, game engines, and Web frameworks [6,7,8]. One main reason for the success of 3DGS, its differentiable Gaussian rasterizer, comes with a significant limitation: its underlying pinhole camera model. This simplification works well as an approximation for many real-world cameras, requiring an additional preprocessing step to undistort the images. However, this undistortion is not well suited for all types of lenses, such as fisheye lenses. Due to their wide Field of View (FoV), they have a high information density and are able to capture scenes with only a few frames. Due to these reasons, they have gained popularity in smartphones and action cameras. The same reasons make fisheye images an interesting candidate for NVS tasks, with a high level of overlap in the images providing depth information for scene reconstruction. Although this does not pose an issue for ray tracing-based methods [2], the underlying 3D Gaussian raster prevents the use of fisheye images with 3DGS. Undistorting these images as pinhole images is possible in principle but usually results in cropping into the images and discarding significant information from the peripheral view, potentially significantly reducing the accuracy of the scene reconstruction. This necessitates the introduction of novel methods that combine the efficiency of 3DGS with the flexibility of using different camera models. Concurrently with our work, an extension of the Gaussian rasterizer for equidistant and panoramic fisheye images, termed Fisheye-GS, has been published [9].

With this contribution, we propose to solve the problem of training a 3DGS model with an arbitrary camera model. We achieve this by using a space-warping module to enable the unmodified pinhole rasterizer to render views for arbitrary optics, including fisheye optics with polynomial distortion, as commonly used to represent real-world cameras [10]. Our proposed space-warping module shifts the scene’s Gaussians according to a predefined distortion function, emulating an arbitrary camera lens. The scale and rotation of the Gaussians are determined as tangential approximations of the actual distortion by leveraging the Jacobian of the distortion function with a subsequent orthogonalization. This approach allows for seamless integration into the existing 3DGS pipeline, as no modifications are made to the rasterization module, enabling the method to work with future versions of the rasterizer. In addition, to reduce floating artifacts in outdoor scenes, we enforce a learnable skybox that is directly integrated into the underlying model of the scene.

This paper is structured as follows. Section 2 reviews the recent literature on NVS methods and their applicability to models without pinhole cameras. Section 3 derives the space-warping module for the Gaussian rasterizer, as well as two explicit distortion functions for the common OpenCV and Blender polynomial fisheye camera models. Section 4 presents the results of extensive experiments on synthetic and real-world datasets, compares our method with other methods, and demonstrates the reasonableness of our design decisions through several ablations, which are subsequently discussed in Section 5, followed by a brief conclusion in Section 6.

## 2. Related Work

In this section, we review the related literature that impacts our research, from traditional Structure from Motion (SfM) to modern methods such as 3DGS. We further position our proposed method among existing methods.

### 2.1. SfM

These methods apply feature-matching algorithms to find correspondences between images. This leads to an initial pose estimation, enabling a dense matching of overlapping views to find a dense depth model of the scene [11,12]. Although classical methods cannot compete with modern learning-based methods with respect to the visual quality of rendered images, their underlying robust pose estimation algorithms are still widely used for the initialization of more recent methods. 3DGS, in particular, relies on the sparse COLMAP model to initialize its Gaussian model for fast convergence [5]. Although there are extensions of learning-based 3D reconstruction methods avoiding the use of classical preprocessing methods for initialization [13,14,15], we consider this beyond the scope of our article and utilize the robust COLMAP pose estimation for initialization of our model.

### 2.2. Neural Radiace Fields (NeRF)

The main contribution accelerating the development of learning-based NVS methods was the introduction of NeRF. Combining an Multilayer Perceptron (MLP) as an implicit differentiable scene representation with a differentiable ray tracer led to a significant step forward in the visual quality of rendered images. In particular, the ability of the underlying MLP to capture view-dependent effects such as reflections enabled a previously unseen level of realism. One major drawback of this method is the inefficiency of the ray tracer and the scene representation, requiring 1–2 days per scene for optimization [2].

Initiated by this contribution, several subsequent papers have attempted to improve upon the initial NeRF. Proposed approaches include anti-aliasing [16], appearance changes [17], deformations [18,19,20], and learned backgrounds [21], the latter being conceptually adopted by our proposed method to reduce floating artifacts introduced by sky textures. Another direction of developments is related to the underlying scene representation. Several papers have successfully demonstrated that the optimization and rendering times can be significantly reduced by choosing a more explicit scene representation, such as octrees [4], feature grids [4], factorized feature planes [19], or hash functions [3]. These explicit methods have achieved real-time rendering speeds and reduced optimization times ranging from hours to minutes [3].

### 2.3. 3DGS

This contribution adopts a similar approach as the preceding explicit methods by representing the scene as a set of 3D Gaussians, each with a 3D position, rotation, 3D scale, opacity, and directional radiance [5]. Three-dimensional Gaussians have been proven to be an efficient graphical primitive with the capability of representing very fine structures. The authors who proposed 3DGS went one step further by simultaneously replacing the previous ray-tracing module with a more efficient differentiable Gaussian rasterizer. Its main benefit is not rendering an image pixel by pixel using individual rays but, instead, rasterizing the whole image at one time. The proposed rasterizer comes with the significant limitation of being restricted to pinhole images, necessitating an undistortion preprocessing step, which prevents the use of wide-angle fisheye images. Eliminating this limitation is the main contribution of our paper by introducing a flexible lightweight space-warping module that enables the emulation of arbitrary camera lenses using the unmodified pinhole rasterizer.

Concurrently with our work, an extension of the Gaussian rasterizer for equidistant and panoramic fisheye images, termed Fisheye-GS, has been published [9]. Our approach conceptually differs in the following aspects: While Fisheye-GS utilizes a modified Gaussian rasterizer, we keep the original rasterizer intact and represent the camera lens using a spatial distortion of the scene without changing. This allows users to implement new camera optics without needing to write additional CUDA code and presents a more modular approach so that the existing rasterizer can be more easily replaced by future versions. In contrast to Fisheye-GS, which focuses on equidistant fisheye images, we derive distortion functions for the common OpenCV polynomial fisheye camera model, eliminating the need for an additional preprocessing step for real-world datasets such as ScanNet++ [22]. In Section 4.2, we provide a qualitative and quantitative comparison between Fisheye-GS and our proposed method.

Another very recently proposed method for scene reconstructions for non-pinhole images is 3D Gaussian ray tracing [23]. Similarly to the initial NeRF, this method relies on a ray-tracing algorithm for rendering, circumventing the issues associated with pinhole rasterization. As is typical for ray-tracing algorithms, this method enables various camera models and effects, such as reflections, refraction, and depth of field. We acknowledge the contribution of the authors; however, we consider 3D Gaussian ray tracing not to be directly comparable to our proposed method, as our main focus lies in enabling non-pinhole optics for the 3DGS rasterization process.

Simultaneously to our submission, a preprint proposing UniGaussian has been published, utilizing a the compression of 3D Gaussians as a similar concept to the warping function introduced in our contribution [24]. Although we utilize an ordered Gram–Schmidt orthogonalization for the computation of the distorted scale and rotation of the 3D Gaussians, UniGaussian relies on an eigendecomposition for this purpose. While acknowledging the work of the authors who proposed UniGaussian, we cannot provide quantitative comparisons with their proposed method, as their source code is not yet publicly available.

## 3. Methods and Data

In this section, we first introduce our approach of emulating arbitrary camera models using a space-warping function. Subsequently, we discuss the specifics of fisheye lenses and, finally, describe our approach of enforcing a learned skybox to reduce floating artifacts. In addition, we describe the synthetic and real-world datasets used in the evaluation of our method.

### 3.1. Space-Warping Function

In 3DGS, a scene is represented by a set of 3D Gaussians. Each Gaussian is fully described by the following properties: position, rotation, scale, opacity, and spherical harmonics coefficients. We denote the original pinhole rendering function as $P_{\mathrm{pinhole}}:R^{3}\to R^{2}$, mapping 3D points to pixel coordinates within the rendered image. As illustrated in Figure 1, we emulate an arbitrary rendering function ($P_{\mathrm{arb}}:R^{3}\to R^{2}$) using a space-warping function ($W:R^{3}\to R^{3}$) that satisfies
(1)$$\begin{array}{c} P_{\mathrm{arb}}\left(x\right)=P_{\mathrm{pinhole}}\left(W(x)\right)\;\forall x\in R^{3}. \end{array}$$

This enables us to keep the underlying rendering function ($P_{\mathrm{pinhole}}$) unchanged while obtaining the desired behavior of $P_{\mathrm{arb}}$. Assuming that W exists, this emulation is correct for the projection of points $x\in R^{3}$. However, distortion of a Gaussian also distorts its scale and rotation, as shown in Figure 2b. As Figure 2c illustrates, we compute the Jacobian, which provides a local linear approximation of the distortion function:(2)$$\begin{array}{c} J_{W}\left(x\right)=\left[\begin{array}{ccc} \frac{\partial W_{1}}{\partial x_{1}} & \frac{\partial W_{1}}{\partial x_{2}} & \frac{\partial W_{1}}{\partial x_{3}}\\ \frac{\partial W_{2}}{\partial x_{1}} & \frac{\partial W_{2}}{\partial x_{2}} & \frac{\partial W_{2}}{\partial x_{3}}\\ \frac{\partial W_{3}}{\partial x_{1}} & \frac{\partial W_{3}}{\partial x_{2}} & \frac{\partial W_{3}}{\partial x_{3}} \end{array}\right]. \end{array}$$

The resulting axes are, in general, not orthogonal, necessitating an additional orthogonalization step. As shown in Figure 2d, for a distorted set of axes ($A\in R^{3\times 3}$), we find an ordering permutation (π) of the axis and compute a Gram–Schmidt orthogonalization of the ordered axes, resulting in orthogonal axes $\tilde{A}\in R^{3\times 3}$:(3)$$\begin{array}{cc} A & =\left[a_{1},a_{2},a_{3}\right], \end{array}$$(4)$$\begin{array}{cc} \pi & \in \left\{\;\sigma \in S_{3}\;|\;|a_{\sigma (1)}\left|\geq \right|a_{\sigma (2)}\left|\geq \right|a_{\sigma (3)}|\;\right\}, \end{array}$$(5)$$\begin{array}{cc} b_{1} & =a_{\pi (1)},\;b_{2}=a_{\pi (2)}-b_{1}\frac{a_{\pi (2)}^{⊤}b_{1}}{b_{1}^{⊤}b_{1}},\;b_{3}=a_{\pi (3)}-b_{1}\frac{a_{\pi (3)}^{⊤}b_{1}}{b_{1}^{⊤}b_{1}}-b_{2}\frac{a_{\pi (3)}^{⊤}b_{2}}{b_{2}^{⊤}b_{2}}, \end{array}$$(6)$$\begin{array}{cc} \tilde{A} & =\left[b_{\pi^{-1}\left(1\right)},b_{\pi^{-1}\left(2\right)},b_{\pi^{-1}\left(3\right)}\right], \end{array}$$
where $S_{3}$ represents the symmetric group, with π being chosen from a constrained set whose elements (σ) all fulfill the ordering constraint.

### 3.2. Fisheye Cameras

Fisheye cameras usually distort the image in relation to the polar angle. For this reason, it is practical to express the warping function (W) in terms of a warping function ($W_{\mathrm{sph}}$) that operates in spherical coordinates. For this purpose, we define the following two transformations: (7)$$\begin{array}{cc} T_{\mathrm{cart}\to \mathrm{sph}}\left(\left[\begin{array}{c} x\\ y\\ z \end{array}\right]\right) & =\left[\begin{array}{c} \sqrt{x^{2}+y^{2}+z^{2}}\\ \arccos\left(z/\sqrt{x^{2}+y^{2}+z^{2}}\right)\\ \arctan\left(y/x\right) \end{array}\right], \end{array}$$(8)$$\begin{array}{cc} T_{\mathrm{sph}\to \mathrm{cart}}\left(\left[\begin{array}{c} r\\ \theta \\ \varphi \end{array}\right]\right) & =\left[\begin{array}{c} r\sin\theta \cos\varphi \\ r\sin\theta \sin\varphi \\ r\cos\theta \end{array}\right]. \end{array}$$

This enables us to write the warping function as
(9)$$\begin{array}{c} W\left(x\right)=\left(T_{\mathrm{sph}\to \mathrm{cart}}\circ D\circ T_{\mathrm{cart}\to \mathrm{sph}}\right)\left(x\right). \end{array}$$

To approximate *D* for a given scene’s camera, we sample a set of corresponding 3D–2D points $\left(X_{i},x_{i}\right)$ and compute their corresponding polar angles as
(10)$$\begin{array}{c} (T_{\mathrm{cart}\to \mathrm{sph}}{\left(X_{i}\right)}_{\theta },T_{\mathrm{cart}\to \mathrm{sph}}{\left(P_{\mathrm{pinhole}}^{-1}\left(x_{i}\right)\right)}_{\theta }). \end{array}$$

As the distortion only depends on the polar angle, we sample these points at equidistant polar angles (θ∈[0,π]) for φ=0 and r=1. Using pairs of polar angles, we fit the coefficients of an 8th-degree polynomial. These coefficients need to be recomputed for any change in the camera model or its specific intrinsic parameters. In our pipeline, this is performed automatically at the start of each optimization run.

With W fully defined by the coordinate transformations and the polynomial polar angle distortion function, we compute their combined Jacobian using the SymPy computer algebra system [25]. The resulting Jacobian is provided in detail in Appendix A. Furthermore, in Appendix B, we demonstrate the application of our proposed method for non-fisheye lenses with an exemplary implementation of an orthographic camera model.

### 3.3. Skybox

In classical SfM, depth can only be computed for points that contain enough features to be matched between different views [11]. For this reason, reconstruction methods tend to struggle with areas with minimal texture, such as uniformly colored walls or skies. This issue can be mitigated by regularizing the depth through the use of pre-learned monocular depth [26] or other prior knowledge about the scene. In our case, reconstructing the sky in open scenes seemed to be most problematic, resulting in inconsistent depth with floating sky artifacts, which reduced the overall rendering quality. We chose to mitigate this issue, similarly to Nerf++ [21], by enforcing a learned skybox at a long distance. We implemented this by placing 1000 isotropic Gaussians on a Fibonacci sphere at a distance twice as far as the furthest point within the sparse initialization point cloud. During optimization, the movement of these Gaussians is restricted to the surface of the initial sphere. In addition, they are prevented from pruning and from having their opacity reset, ensuring a consistent background being learned during optimization. Compared to other methods [21], instead of creating a separate model for the background, we fully integrate the background Gaussians into a single model, enabling a seamless rendering process.

### 3.4. Datasets

We decided to evaluate our proposed method on two different datasets—one synthetic and one real-world dataset. The synthetic dataset (Blender) eliminates most sources of errors and allows us to focus solely on the performance of the model under perfect conditions, while the real-world dataset (ScanNet++) enables comparisons with existing methods and provides convincing indications about the applicability of the method.

#### 3.4.1. Synthetic Blender Dataset

We chose 6 publicly available blender scenes containing 3 indoor and 3 outdoor scenes [27]:Archiviz;Barbershop;Classroom;Monk;Pabellon;Sky.

For each scene, we rendered 100 photorealistic, ray-traced frames along a predefined trajectory with a resolution of 1024×1024px. Each 8th image was taken as a test image; all other images were part of the training set. We chose a simulated fisheye camera with an FoV of 180° and a sensor size of 32mm. Blender’s camera model is configured by specifying five coefficients, which were chosen with the following values [28]:$$\begin{array}{c} (1.0\;\cdot \;10^{-5},\;-8.7\;\cdot \;10^{-2},\;-3.5\;\cdot \;10^{-6},\;3.5\;\cdot \;10^{-6},\;-2.6\;\cdot \;10^{-8}). \end{array}$$

In Blender’s camera model, these coefficients model a polynomial mapping radial distances on the camera sensor to camera rays. The values for the coefficients were chosen as a large negative linear component to achieve fisheye distortion with a large FoV. Furthermore, we chose small coefficients for higher orders to avoid artifacts caused by a non-injective distortion function while avoiding zero-valued coefficients to preserve the projection’s complexity. For each scene, we computed our own polynomial representation of the distortion function according to the samples presented in Equation (Equation 10).

#### 3.4.2. ScanNet++ Dataset

This dataset is provided by the Technical University of Munich and consists of several indoor scenes with images, camera poses, and point clouds [22]. For comparability with Fisheye-GS, we chose the following 6 scenes:Bedroom (e8ea9b4da8);Kitchen (bb87c292ad);Office Day (4ba22fa7e4; the corresponding scene from the Fisheye-GS paper is currently not available, so we chose a similar scene);Office Night (8d563fc2cc);Tool Room (d415cc449b);Utility Room (0a5c013435).

Each scene consists of a varying number of fisheye images (147–406) with a resolution of 1752×1168px. Again, each 8th image was taken as a test image; all other images were part of the training set. For each scene, the authors provide the COLMAP poses, intrinsics, and a sparse point cloud. For the Blender dataset, we converted each scene’s intrinsic parameters to our custom distortion function using the samples described in Equation (Equation 10).

## 4. Results

This section contains the results of our proposed method for synthetic and real-world datasets, as well as comparisons with Fisheye-GS. We further show several ablations to validate the design choices of our method. The experiments were carried out on a system with an *AMD EPYC 72F3*, 256GB of memory, and an NIVIA A100 GPU. However, the overall system requirements are lower, identical to 3DGS.

### 4.1. Synthetic Blender

As a first experiment, we ran our method on the six synthetic blender scenes for 30,000 iterations. The per-scene results, reporting the number of Gaussians, Peak Signal-to-Noise Ratio (PSNR), Structural Similarity Index Measure (SSIM) [29], and Learned Perceptual Image Patch Similarity (LPIPS) [30], are shown in Table 1. As indicated by the metrics and the qualitative samples in Figure 3, Figure 4 and Figure 5, the rendered images achieve photorealistic quality with only minor artifacts. As seen in the rendered *Barbershop* image, the method struggles with reflections in the mirror. This is expected because the spherical harmonics do not provide sufficient capacity for encoding the full reflected room within the Gaussians of the mirror. A detailed illustration of this problem is shown in Figure 3.

The rendered images clearly show a high level of local reconstruction error in the region of the mirror. As the rendered depth shows, within the mirror, there are Gaussians on the mirror’s surface, as well as Gaussians behind the mirror. Due to their limited expressiveness, the Gaussians on the mirror’s surface lead to blurry horizontal artifacts. The other Gaussians represent a mirrored room behind the mirror, which is heavily under-sampled, preventing a photorealistic reconstruction. According to our understanding, this issue is inherent to 3DGS, as it does not model any secondary rays. Furthermore, there are minor artifacts visible in the shady regions of the *Monk* scene. An outlier is the *Pabellon* scene, which shows significantly worse metrics than all other methods. In this scene, the method struggles to render the texture of the pool, causing blurry artifacts.

### 4.2. ScanNet++

To further confirm the synthetic results, we performed a second experiment on the real-world Scannet++ dataset for 30,000 iterations, using our proposed method and Fisheye-GS. As the authors who proposed Fisheye-GS thoroughly demonstrated, the use of classical 3DGS for the reconstruction of fisheye scenes performs significantly worse than Fisheye-GS, so in this paper, we did not compare our method against 3DGS and relied on Fisheye-GS as a benchmark. We did not optimize a separate skybox in this experiment because of the absence of outdoor scenes. As our method is directly applicable to OpenCV fisheye images, the pre-processing step to convert the images to equidistant projections was only performed for Fisheye-GS. In Table 2, we report the number of Gaussians, PSNR, SSIM, and LPIPS for each experimental run. For better comparability, we also report the mean of the normalized differences for each metric. Qualitative samples are shown in Figure 6 and Figure 7. When comparing the mean results, our method outperforms Fisheye-GS on every metric except SSIM. However, the increases in PSNR and LPIPS are relatively small, and the main improvement of our method over Fisheye-GS is the reduction in the number of Gaussians and, thus, the model size, which shows a relative difference of 42.18%.

### 4.3. Ablations

To test the design decisions for our proposed method, we chose to conduct experiments to validate the Jacobian distortion function, the learned skybox, and the number of coefficients used for the polar distortion polynomial.

#### 4.3.1. Jacobian Distortion

To qualitatively visualize the importance of this design component, we created a scene consisting of Gaussians along the edges of a cube. Each Gaussian was stretched along the axis of its corresponding edge. As Figure 8 shows, disabling Jacobian distortion leads to blurred edges of the cube, since the Gaussians are not rotated according to the spatial distortion. To confirm this observation quantitatively, we performed an optimization on the *Utility Room* scene, showing overall improved visual metrics with the Jacobian distortion enabled, as shown in Table 3.

#### 4.3.2. Degree of the Distortion Polynomial

To test the effect of varying the number of polynomial coefficients within the polar distortion function, we performed experiments on the *Utility Room* scene with an increasing number of coefficients. As shown in Table 4 and Figure 9, all metrics improve with an increasing number of coefficients, with only minor improvements from sixth degree upward. This is to be expected, as increasing the number of coefficients decreases the approximation error, with diminishing returns when the approximation error approaches the pixel size of the rendered image. For the chosen polynomial degree of eight, we computed the sensitivity as the mean absolute deviation from the results for higher and lower polynomial degrees and found the values of $\Delta \mathrm{PSNR}=1.29\;\cdot \;10^{-1}$, $\Delta \mathrm{SSIM}=1.75\;\cdot \;10^{-4}$, and $\Delta \mathrm{LPIPS}=1.97\;\cdot \;10^{-4}$. This demonstrates the overall robustness of the proposed method for the chosen polynomial degree.

#### 4.3.3. Learnable Skybox

For validation of the learned skybox, we performed two optimizations on the synthetic outdoor *Monk* scene—one with the skybox enabled and one with it disabled. As Table 5 shows, all visual metrics improve significantly with the enabled feature. In addition, Figure 10 illustrates the reason for this discrepancy. With the learned skybox disabled, the sky is represented using a set of Gaussians in close proximity to the camera, which creates floating sky artifacts and hides parts of the scene. The enforcement of the learned skybox successfully mitigates this issue.

### 4.4. Performance

To measure the performance of the model, we rendered 100 random views for each model when trained on the ScanNet++ dataset and measured the inference latency. As Table 6 shows, the current version of our method achieves interactive frame rates for most scenes, whereas it shows high latencies for the *Tool Room* scene. This scene contains significantly more Gaussians than the others, indicating a possible bottleneck for rendering speeds. This behavior is expected, as our space-warping module leads to an additional computational effort linear to the number of Gaussians being processed. We acknowledge the existence of several projects focusing on the acceleration of 3DGS [8,31]; however, we consider this process of optimization to be outside of the scope of this paper.

## 5. Discussion

With the photorealistic synthetic rendering results, we can confirm that our method is a valid approach for extending 3DGS to arbitrary camera models. Although there are still some reconstruction artifacts in shaded parts of the scene and highly reflective surfaces, the metrics indicate a high visual quality. Furthermore, despite a significant discrepancy between synthetic and real-world results, our method matched or outperformed the Fisheye-GS method on the real-world dataset. From a quantitative viewpoint, the most significant improvement of our space-warping approach is the reduction in the number of Gaussians, enhancing the overall space efficiency of the reconstructed model. Qualitatively, our method processes ScanNet++ OpenCV fisheye images directly, removing the need for additional preprocessing steps. Additionally, extensive ablations supported the design decisions in our method, such as the learned skybox, the Jacobian distortion, and the number of polynomial coefficients. The learned skybox, in particular, effectively mitigates floating artifacts and improves the overall visual quality of outdoor scenes.

However, several open questions remain. One is the generalization capability of our model. To compare our method with Fisheye-GS, we adopted the same eighth image selection strategy for the test set. To further explore generalization, future research could include rendering views from more distinct viewpoints or depth comparisons with ground-truth data. Another limitation is the real-time capability of our model. While it achieves interactive frame rates for most scenes, performance in larger scenes suggests a need for further optimizations to reduce rendering latency.

## 6. Conclusions and Outlook

Our method demonstrates significant advancements in extending 3DGS to arbitrary camera models, particularly in terms of space efficiency and preprocessing requirements. The ability to process fisheye images without additional steps and the improved visual quality achieved through the learned skybox showcase the potential of our approach. Future research directions include optimizing camera parameters to reduce the reliance on robust and precise calibration processes. The trainability of polynomial lens distortion coefficients could be explored to enhance adaptability. Building on the modularity of our method, it could integrate seamlessly with existing extensions for 3DGS, such as for dynamic scene reconstruction [32], reconstruction without known poses [15], or more compact scene representations [33]. Finally, we are currently developing the warping module as a *gsplat* component. This step may enable the integration of our method into real-time applications, broadening its practical utility and impact.

## Figures and Tables

**Figure 1 jimaging-10-00330-f001:**
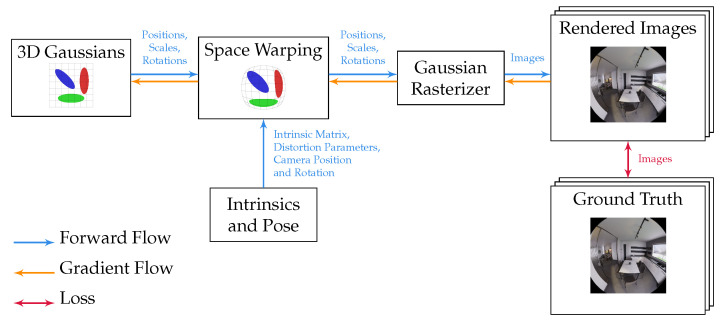
Pipeline of our proposed method. Before being forwarded to the pinhole Gaussian rasterizer, we apply a space-warping module to the position, rotation, and scale to emulate the distortion of a lens specified by the camera’s intrinsics.

**Figure 2 jimaging-10-00330-f002:**
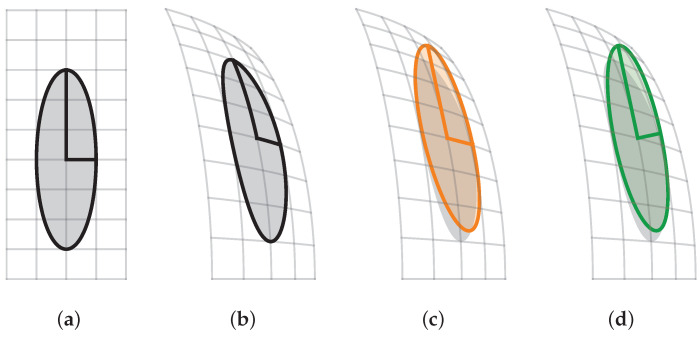
Distortion of scale and rotation. The four images show the steps in the distortion pipeline from left to right. An undistorted Gaussian (**a**) is non-linearly distorted (**b**). This distortion is linearly approximated using Jacobian $J_{W}$ (**c**), with a subsequent orthogonalization of the axes (**d**). For (**c**,**d**), the gray area shows the true distorted Gaussian to visualize the approximation error.

**Figure 3 jimaging-10-00330-f003:**
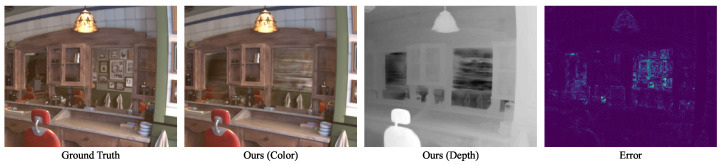
Reconstruction results for the synthetic *Classroom* scene.

**Figure 4 jimaging-10-00330-f004:**
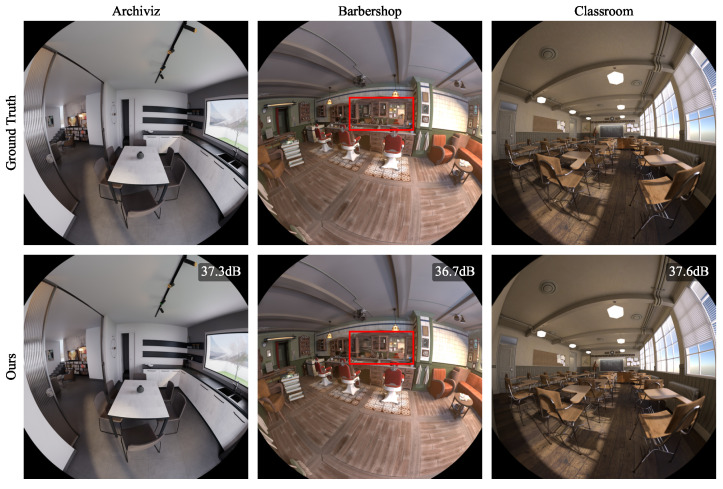
Results of our proposed method on synthetic Blender scenes (*Archiviz*, *Barbershop*, and *Classroom*). Red rectangles indicate areas in which our method produced reconstruction artifacts. Zoom in for details.

**Figure 5 jimaging-10-00330-f005:**
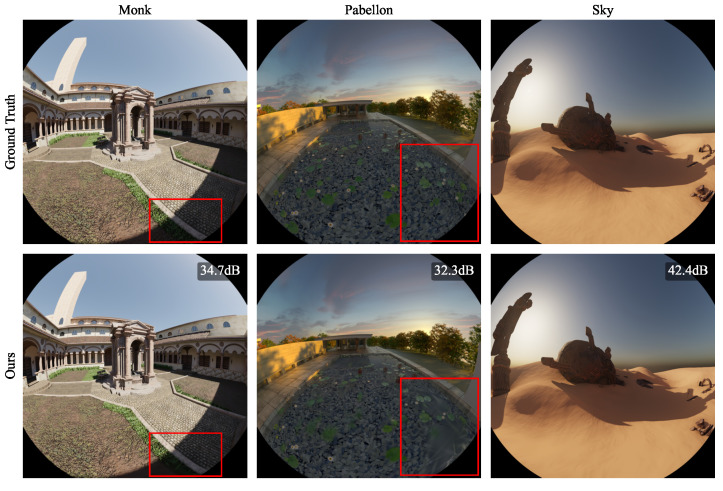
Results of our proposed method on synthetic Blender scenes (*Monk*, *Pabellon*, and *Sky*). Red rectangles indicate areas in which our method produced reconstruction artifacts. Zoom in for details.

**Figure 6 jimaging-10-00330-f006:**
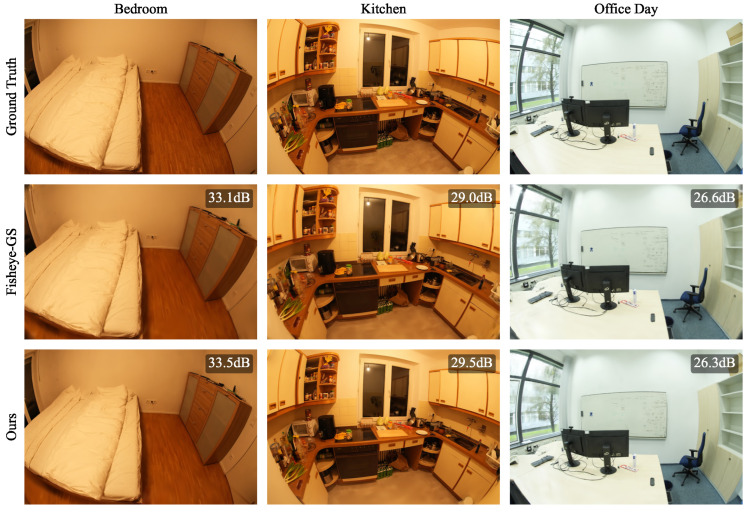
Results of our proposed method and Fisheye-GS on ScanNet++ scenes (*Bedroom*, *Kitchen*, and *Office Day*). Zoom in for details.

**Figure 7 jimaging-10-00330-f007:**
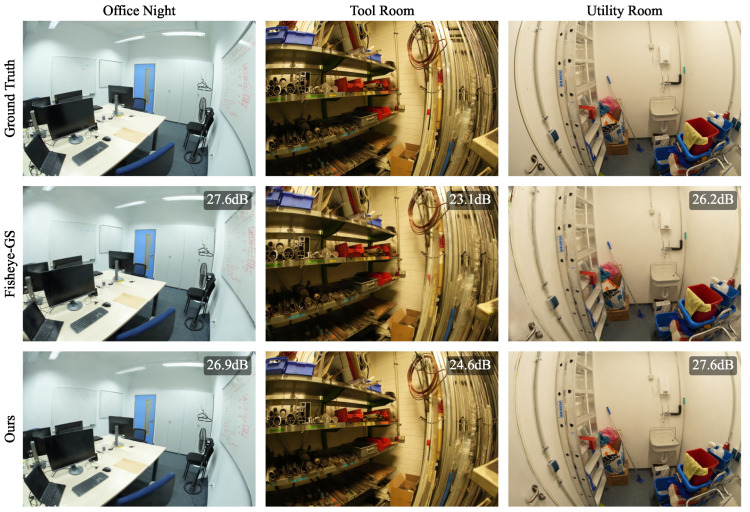
Results of our proposed method and Fisheye-GS on ScanNet++ scenes (*Office Night*, *Tool Room*, and *Utility Room*). Zoom in for details.

**Figure 8 jimaging-10-00330-f008:**
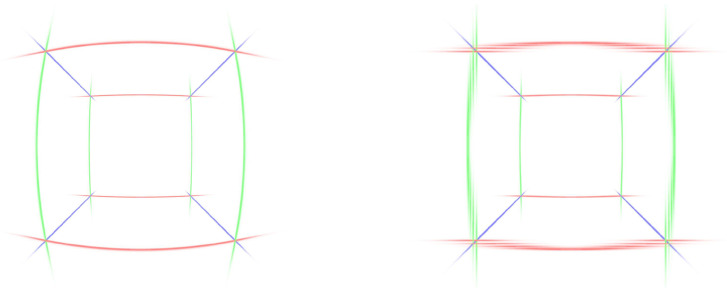
Renderings of a cube with Gaussians along the edges. The left rendering has the scale and rotation adjusted according to the Jacobian; for the right rendering, scale and rotation were left unmodified.

**Figure 9 jimaging-10-00330-f009:**
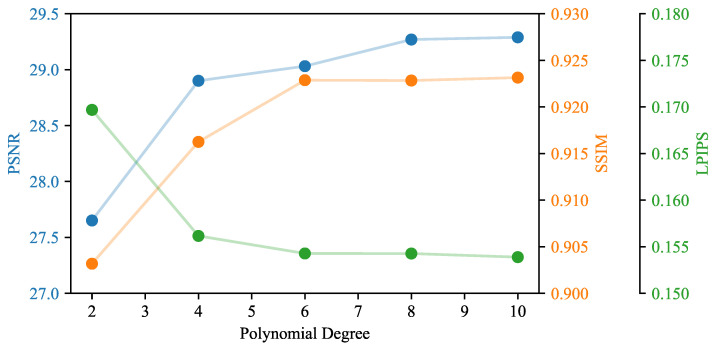
Evaluation metrics for the *Utility Room* scene for varying degrees of the polynomial polar distortion function.

**Figure 10 jimaging-10-00330-f010:**
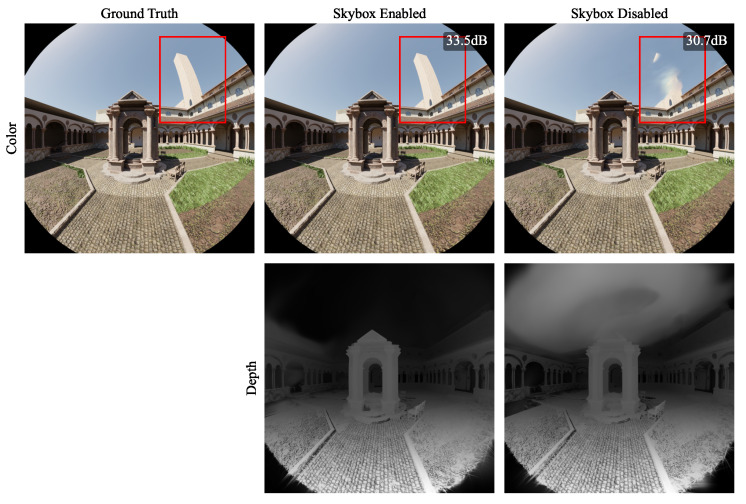
Results for our proposed model trained on synthetic data with the learned skybox enabled (middle) and disabled (right).

**Table 1 jimaging-10-00330-t001:** Experimental results of the proposed method on the synthetic Blender dataset. For each metric, the arrows indicate if lower or higher results are preferred.

Scene	#Gaussians↓	PSNR↑	SSIM↑	LPIPS↓
Archiviz	604,504	38.36	0.979	0.064
Barbershop	573,016	36.97	0.979	0.048
Classroom	676,173	35.56	0.972	0.094
Monk	470,918	33.74	0.964	0.061
Pabellon	502,938	33.91	0.910	0.209
Sky	118,730	42.29	0.989	0.036

**Table 2 jimaging-10-00330-t002:** Experimental results of the proposed method on the ScanNet++ dataset. For each dataset, the best result is highlighted in bold.

Scene	Method	#Gaussians↓	PSNR↑	SSIM↑	LPIPS↓
Bedroom	Fisheye-GS	345,084	32.09	**0.947**	0.192
	Ours	**239,923**	32.09	0.946	**0.191**
Kitchen	Fisheye-GS	593,270	30.75	**0.935**	**0.179**
	Ours	**392,722**	**30.80**	0.928	0.188
Office Day	Fisheye-GS	669,351	25.92	**0.888**	0.182
	Ours	**480,055**	**26.22**	0.885	**0.176**
Office Night	Fisheye-GS	684,649	**26.30**	**0.907**	**0.176**
	Ours	**448,746**	26.27	0.899	0.179
Tool Room	Fisheye-GS	2,647,082	**27.01**	**0.856**	0.222
	Ours	**1,479,128**	26.95	0.851	**0.221**
Utility Room	Fisheye-GS	749,024	28.06	0.915	0.165
	Ours	**453,970**	**29.20**	**0.923**	**0.155**
Relative	Fisheye-GS	42.81%	−0.83%	**0.31%**	0.76%
Mean	Ours	**−42.81%**	**0.83%**	−0.31%	**−0.76%**

**Table 3 jimaging-10-00330-t003:** Experimental results for evaluation of the Jacobian distortion for rotation and scale of the Gaussians.

Scene	Jacobian	#Gaussians↓	PSNR↑	SSIM↑	LPIPS↓
Utility Room	Enabled	455,653	**29.07**	**0.923**	**0.155**
	Disabled	**387,457**	28.80	0.918	0.171

**Table 4 jimaging-10-00330-t004:** Experimental results for evaluation of the number of polynomial coefficients in the polar distortion polynomial.

Scene	Degree	#Gaussians↓	PSNR↑	SSIM↑	LPIPS↓
Utility Room	2	537,376	27.65	0.903	0.170
	4	665,298	28.90	0.916	0.156
	6	472,092	29.03	**0.923**	**0.154**
	8	**456,368**	29.27	**0.923**	**0.154**
	10	456,941	**29.29**	**0.923**	**0.154**

**Table 5 jimaging-10-00330-t005:** Experimental results for evaluation of the learned skybox.

Scene	Skybox	#Gaussians↓	PSNR↑	SSIM↑	LPIPS↓
Monk	Disabled	**389,608**	31.92	0.958	0.074
	Enabled	471,256	**33.63**	**0.963**	**0.062**

**Table 6 jimaging-10-00330-t006:** Average rendering latencies for our proposed model when trained on the ScanNet++ dataset.

Scene	Latency/ms	#Gaussians
Bedroom	27.4±3.5	239,923
Kitchen	33.8±4.8	392,722
Office Day	35.9±3.5	480,055
Office Night	38.2±3.6	448,746
Tool Room	120.8±18.9	1,479,128
Utility Room	37.9±5.3	453,970

## Data Availability

We provide all source code and the synthetic blender dataset in the project’s GitHub repository: https://github.com/jna-358/warped-gaussians, accessed on 16 December 2024. The ScanNet++ dataset requires an application submitted through its publisher’s website for download (https://kaldir.vc.in.tum.de/scannetpp, accessed on 16 December 2024).

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
