# Peer review of "Arbitrary Optics for Gaussian Splatting Using Space Warping"

_2313-433X, 2024, doi:10.3390/jimaging10120330_

Round 1

Reviewer 1 Report

Comments and Suggestions for Authors

The paper effectively identifies the limitations of existing 3D Gaussian Splatting (3DGS) methods and introduces a well-conceived novel approach to address them. The proposed space warping function and the integration of a learned skybox are innovative and represent significant advancements in the field. The authors have provided comprehensive validation of their method through experiments on both synthetic and real-world datasets, as well as through detailed ablation studies. These efforts convincingly demonstrate the method's effectiveness.

I recommend the paper for acceptance with minor revisions:

The provided link to the codebase is currently inactive. Please correctly point readers to the repository.

In Equations 7 and 8, the second component on the right-hand side might require clarification. Specifically, the spherical coordinate transformation is typically defined as: arccos(z/sqrt(x^2+y^2+z^2) for the second component of Equation 7, and r*sin(theta)*sin(phi) for the second component of Equation 8. If these differences are intentional, please clarify. 

Author Response

Dear reviewer, thank you for providing your constructive feedback on our initial manuscript. We have made an effort to address each of the comments thoroughly in the revised version of the paper. For a detailed overview of the changes, we append a modified version of the revised paper with all changes highlighted in as red text. We believe that the revisions have significantly improved the manuscript and we appreciate your contribution in this process. 

For a more detailed response to your comments, refer to the section below.

  1. The provided link to the codebase is currently inactive. Please correctly point readers to the repository.

    The repository link had been unintentionally set to private for the first few days of the review process. It should now be publicly available.

  2. In Equations 7 and 8, the second component on the right-hand side might require clarification. Specifically, the spherical coordinate transformation is typically defined as: arccos(z/sqrt(x^2+y^2+z^2) for the second component of Equation 7, and r*sin(theta)sin(phi) for the second component of Equation 8. If these differences are intentional, please clarify.

    Your comment is correct, the error in the manuscript has been fixed accordingly. All implementations had been done with the correct definition of spherical coordinates.

Reviewer 2 Report

Comments and Suggestions for Authors

This paper introduces a novel method for extending 3D Gaussian splatting (3DGS) to accommodate photorealistic rendering to arbitrary camera optics, including highly distorting fish-eye lenses with polynomial distortion. The underlying idea is based on incorporating a differentiable space warping model that adjusts the scene's Gaussian representation to emulate different lens distortions, overcoming limitations that arise under the traditional pinhole camera model. A learnable sky-box is also added to improve scene reconstruction accuracy of textureless objects and skyes, particularly in outdoors settings. Experimental results are presented, using six available Blender synthetic datasets, and also with real-world datasets consisting on six different scenes. Results are compared in performance to existing similar or related methods, and design choices are validated through ablation studies. Finally, the authors discuss potential future research directions for enhancing the model's capabilities.

Author Response

Dear reviewer, thank you for providing your constructive feedback on our initial manuscript. We have made an effort to address each of the comments thoroughly in the revised version of the paper. For a detailed overview of the changes, we append a modified version of the revised paper with all changes highlighted as red text. We believe that the revisions have significantly improved the manuscript and we appreciate your contribution in this process. 

For a more detailed response to your comments, refer to the section below.

  1. Improvements in research design

    The revised version of the paper contains several improvements in the overall research design: To evaluate the generalization capabilities of the method, we evaluated it on five additional scenes from the scannet dataset, selected by the diversity of shown objects. We additionally demonstrated the method’s flexibility by implementing and testing an orthographic camera model.

  2. Improvements in method description

    To improve the method’s description, we introduced a glossary of acronyms, improved the presentation of the ablation studies with an additional graph, and added a new figure illustrating reflection artifacts in detail.

Reviewer 3 Report

Comments and Suggestions for Authors

The paper is technically sound, as it builds on established methods such as 3DGS and COLMAP while introducing well-justified improvements. The mathematical formulation of the space-warping function and its integration into the 3DGS pipeline are rigorously defined. Furthermore, the experimental results validate the effectiveness of the proposed method, demonstrating improvements in both synthetic and real-world datasets. This interesting topic is within the scope of the Imaging Journal.  However, I have a few comments. Please refer to them.

 1) The paper could benefit from a broader range of real-world datasets to further validate its generalizability.

2) The impact of different distortion models could be further explored, as the current study primarily focuses on polynomial distortions.

3) A more detailed comparison with other non-pinhole approaches could strengthen the motivation for the proposed method.

4) Including a sensitivity analysis of hyperparameters (e.g., polynomial degree) would add value.

5) The paper could provide more detailed future directions, such as potential integration with real-time applications or other camera types. How does the proposed space-warping module compare to existing alternatives regarding computational complexity and flexibility? Could its implementation be extended to different camera models?

6) What potential biases or limitations exist in the synthetic and real-world datasets? How might these impact the generalizability of the results?

7) Are the ablation studies (e.g., impact of Jacobian distortion, polynomial degree, and learnable skybox) sufficient to isolate the contributions of each component?

8) How are the observed artifacts (e.g., mirror reflections and pool texture blurriness) analyzed? Are these issues inherent to the method or specific to certain scenes?

9) Are the chosen metrics (PSNR, SSIM, LPIPS) appropriate for evaluating photorealistic quality and structural fidelity? Are there additional metrics that could provide deeper insights?

10) References must be formatted according to the Imaging template

Author Response

Dear reviewer, thank you for providing your constructive feedback on our initial manuscript. We have made an effort to address each of the comments thoroughly in the revised version of the paper. For a detailed overview of the changes, we append a modified version of the revised paper with all changes highlighted as red text. We believe that the revisions have significantly improved the manuscript and we appreciate your contribution in this process. 

For a more detailed response to your comments, refer to the section below.

  1. The paper could benefit from a broader range of real-world datasets to further validate its generalizability.
    To further validate the generalizability of our method, we selected five additional scenes from scannet dataset and added their reconstruction results to the appendix of our paper. We chose the five scenes to be dissimilar to the five main scenes from our evaluation in order to cover a broader range of reconstructed objects:
    • Printer: 1b9692f0c7
    • Conference Room: 1b75758486
    • Electricity Room: 1c4b893630
    • Bathtub: 1c876c250f
    • Plant: 06a3d79b68
  2. The impact of different distortion models could be further explored, as the current study primarily focuses on polynomial distortions.

    To demonstrate the generalizability with respect to non-polynomial camera models, we added the implementation and evaluation for orthographic camera models. For evaluating this camera model, we created an additional synthetic dataset and demonstrate photorealistic reconstruction quality on this dataset.

  3. A more detailed comparison with other non-pinhole approaches could strengthen the motivation for the proposed method.

    To the author’s knowledge, at the time of our research there were no significant non-pinhole contributions in the field of 3DGS aside from the mentioned Fisheye-GS paper. Since then, there has been published the preprint article UniGaussian by Ren et al. at 22 Nov 2024 that utilizes a very similar approach to our method with a focus on urban scene representation. Due to its recency, the source code is not yet publicly available, preventing quantitative comparisons at this stage. Another very recent contribution 3D Gaussian Ray Tracing by Monenne-Loccoz (current version from 10 Oct 2024) implements non-pinhole optics for Gaussian scene representations, but implements a whole new rendering raytracing rendering pipeline, circumventing the rasterization process. Our target is enabling non-pinhole camera models for rasterization, thus we consider their contribution to not be directly comparable to ours.

    To acknowledge these very recent contributions and strengthen the motivation of our proposed method, we added a section that discusses those method within the related work section of the revised paper.

  4. Including a sensitivity analysis of hyperparameters (e.g., polynomial degree) would add value.

    For better visualization, we added a figure illustrating the metrics under varying values for the polynomial degree. We further added sensitivity values for each metric for the chosen polynomial degree of 8, demonstrating the robustness of this choice.

  5. The paper could provide more detailed future directions, such as potential integration with real-time applications or other camera types. How does the proposed space-warping module compare to existing alternatives regarding computational complexity and flexibility? Could its implementation be extended to different camera models?

    For more detailed future directions, in the revised version of the paper, we added further elaborations within the final section regarding our current work in the implementation of real-time capable applications. Further, we demonstrated the simplicity of the integration of other camera types and the overall flexibility of our approach with the implementation of an orthographic camera. Regarding the computational complexity, we added a short sentence within the latency analysis section discussing the additional computations which are linear in the number of Gaussians.

  6. What potential biases or limitations exist in the synthetic and real-world datasets? How might these impact the generalizability of the results?

    We acknowledge potential biases in the selection process of the synthetic blender scenes. As these synthetic scenes were only used to demonstrate the feasibility of our method under “perfect” conditions, we do not see any negative impact of potential biases here. Another potential bias lies within the scenes selected by the authors of “Fisheye-GS”, which are used by us for comparison purposes. To tackle this issue, we included an additional five scenes in the appendix of the paper, ensuring a broader coverage of represented scenes.

  7. Are the ablation studies (e.g., impact of Jacobian distortion, polynomial degree, and learnable skybox) sufficient to isolate the contributions of each component?

    We conducted isolated ablation studies with the only modified component being the component of interest (such as jacobian distortion, skybox, or polynomial degree). For each component, we demonstrated that removing it from the overall pipeline decreased the reconstruction results. We consider these experiments to be sufficient to isolate the contribution of each component. In the revised version of the paper, we conducted an additional sensitivity analysis for the polynomial coefficients, providing a deeper insight into the contribution of the distortion component.

  8. How are the observed artifacts (e.g., mirror reflections and pool texture blurriness) analyzed? Are these issues inherent to the method or specific to certain scenes?

    The artifacts are detected as image regions with a high local deviation from the ground truth image. To further discuss these artifacts, we added a detailed illustration and discussion for the reflection artifacts in the barbershop scene. To our understanding, these artifacts are inherent to the class of SfM problems, as there is an inherent ambiguity in the reconstruction of reflective surfaces.

  9. Are the chosen metrics (PSNR, SSIM, LPIPS) appropriate for evaluating photorealistic quality and structural fidelity? Are there additional metrics that could provide deeper insights?

    The three metrics used in the paper are the de-facto standard within the research domain of novel view synthesis, as illustrated by the following seminal papers:
        - 3D Gaussian Splatting for Real-Time Radiance Field Rendering (Kerbl et al.) [5]
        - NeRF: Representing Scenes as Neural Radiance Fields for View Synthesis (Mildenhall et al.) [2]
        - Mip-NeRF: A Multiscale Representation for Anti-Aliasing Neural Radiance Fields (Barron et al.) [12]
    We acknowledge the existence of other pixel level metrics (RSME, L1 etc.) as well as perceptual metrics (NLPD, FID, etc.), but do not see a significant improvement by their inclusion due to the lack of comparability with existing methods in this field.

  10. References must be formatted according to the Imaging template

    We adhere to the official MDPI template at https://www.overleaf.com/latex/templates/mdpi-article-template/fcpwsspfzsph and coordinate any modifications with the editors of J.Imaging.

Reviewer 4 Report

Comments and Suggestions for Authors

This paper suggests an extension to an existing method for scene reconstruction, addressing its limitation of relying on the pinhole camera model. By introducing a differentiable warping function to the Gaussian scene representation, the authors' approach enables accurate scene reconstruction using arbitrary camera optics, including highly distorting fisheye lenses.

The paper is nice and I enjoyed reading it; however, I have several concerns:

1. The paper needs a table of acronyms to facilitate the search for their definition. Some acronyms may have more than one definition, so it is necessary to have a consistency of the terms.

2. In Figure 1, the author needs to specify what information goes through each arrow drawn in the figure.

3. In equation 4, what does Sigma denote?

4. In line 152, the authors write "we sample a set of corresponding 3D-2D points (Xi, xi)". How do the authors determine the specific points to be sampled?

5. In line 195, the authors write "configured by specifying five coefficients, which have been chosen as the following values:"; however, no explanation is given why these coefficients have been chosen.

6. The comparison made in Table 2 is very important; however, why did the authors put it in a table? The data would be easier to understand if it were presented in a graph.

7. In Morgenstern, W., Barthel, F., Hilsmann, A., & Eisert, P. (2025), "Compact 3d scene representation via self-organizing gaussian grids", In European Conference on Computer Vision (pp. 18-34). Springer, Cham., the authors suggest a compact scene representation organizing the parameters of 3D Gaussian Splatting (3DGS) into a 2D grid with local homogeneity, ensuring a drastic reduction in storage requirements. This technique can be very useful for the suggested model.

8.  There is also an approach to compensate for the need of images with complex spatial patterns by many basic uncomplicated images. In Wiseman, Y. (2010, July), "Take a picture of your tire!. In Proceedings of 2010", IEEE International Conference on Vehicular Electronics and Safety, pp. 151-156. available online at: https://citeseerx.ist.psu.edu/document?repid=rep1&type=pdf&doi=4fe6531508aebdfcb120309118b6ad034e5f2ca0  the author suggests rolling the tire and take many picture of it in all of its parts in order to find damages. I would encourage the author to cite this paper and explain why they prefer another attitude.

9. The title of section 5 should be changed to "Conclusions and Future Work".

10. The format of references should be consistent.

Author Response

Dear reviewer, thank you for providing your constructive feedback on our initial manuscript. We have made an effort to address each of the comments thoroughly in the revised version of the paper. For a detailed overview of the changes, we append a modified version of the revised paper with all changes highlighted as red text. We believe that the revisions have significantly improved the manuscript and we appreciate your contribution in this process. 

For a more detailed response to your comments, refer to the section below.

  1. The paper needs a table of acronyms to facilitate the search for their definition. Some acronyms may have more than one definition, so it is necessary to have a consistency of the terms.

    A glossary with all abbreviations used in the paper has been added.

  2. In Figure 1, the author needs to specify what information goes through each arrow drawn in the figure.

    In the revised paper, we added labels to the arrows describing the type of information being processed in the corresponding step of the pipeline.

  3. In equation 4, what does Sigma denote?

    Sigma is an auxiliary variable used to constrain the symmetric group to only those permutations which fulfill the ordering constraint. A clarification has been added to the revised paper.

  4. In line 152, the authors write "we sample a set of corresponding 3D-2D points (Xi, xi)". How do the authors determine the specific points to be sampled?

    The points are sampled along a semicircle at equidistant polar angles for an azimuth angle of zero and a unit radius. This description has been added to the revised paper.
  5. In line 195, the authors write "configured by specifying five coefficients, which have been chosen as the following values:"; however, no explanation is given why these coefficients have been chosen.

    We manually chose these parameters to model a highly distorted fisheye lens with a large field of view. To prevent a drastic simplification of the problem, we ensured to have non-zero coefficients at every degree of blender’s distortion polynomial. Further, to avoid a non-injective distortion function, we needed to ensure low contributions at high degrees of the polynomial, otherwise non-realistic distortion artifacts would be introduced with the image folding into itself. Further, within the blender user interface, the coefficients are specified in degrees, while the internal representation is in radians, causing seemingly odd coefficient values. For clarification, we added a short paragraph to the revised paper.

  6. The comparison made in Table 2 is very important; however, why did the authors put it in a table? The data would be easier to understand if it were presented in a graph.

    We acknowledge the advantage of figures to provide a more intuitive understanding of quantitative results than reporting raw metrics. For Table 4, we applied this principle and added a figure to better visualize the relation between the reconstruction results and increasing polynomial degrees.

    Table 2, however, reports per-dataset reconstruction scores for our proposed method and Fisheye-GS. We do not see any second dimension that these scores could be put in relation to within a graph. One-dimensional plots (such as box-plots) could be used to compare the statistical distribution for each method over all datasets. However, the scores are highly dependent on the datasets themselves, skewing such forms of presentation heavily towards datasets with high scores. For this reason, reporting the per-dataset metrics for each method in combination with the presentation of representative image samples is the de-facto standard for reporting results in the field of novel-view-synthesis, as illustrated by the following publications:
        - NeRF: Representing Scenes as Neural Radiance Fields for View Synthesis (Mildenhall et al.)
        - 3D Gaussian Splatting for Real-Time Radiance Field Rendering (Kerbl et al.)
        - Compact 3D Scene Representation via Self-Organizing Gaussian Grids (Morgenstern et al.)
        - Fisheye-GS: Lightweight and Extensible Gaussian Splatting Module for Fisheye Cameras (Liao et al.)
        - Mip-NeRF: A Multiscale Representation for Anti-Aliasing Neural Radiance Fields (Barron et al.)

  7. In Morgenstern, W., Barthel, F., Hilsmann, A., & Eisert, P. (2025), "Compact 3d scene representation via self-organizing gaussian grids", In European Conference on Computer Vision (pp. 18-34). Springer, Cham.,‏ the authors suggest a compact scene representation organizing the parameters of 3D Gaussian Splatting (3DGS) into a 2D grid with local homogeneity, ensuring a drastic reduction in storage requirements. This technique can be very useful for the suggested model.

    We consider this to be an interesting research direction and added it to the outlook of the revised paper.
  8. There is also an approach to compensate for the need of images with complex spatial patterns by many basic uncomplicated images. In Wiseman, Y. (2010, July), "Take a picture of your tire!. In Proceedings of 2010", IEEE International Conference on Vehicular Electronics and Safety, pp. 151-156. available online at: https://citeseerx.ist.psu.edu/document?repid=rep1&type=pdf&doi=4fe6531508aebdfcb120309118b6ad034e5f2ca0 the author suggests rolling the tire and take many picture of it in all of its parts in order to find damages. I would encourage the author to cite this paper and explain why they prefer another attitude.

    Our paper proposes a modification of the state of the art 3D reconstruction method 3DGS for arbitrary camera models. However, the suggested paper deals with tire damage detection using the JPEG compression algorithm, which is not directly related to our research topic. The scientific issue they addressed is not directly relevant to our research. Therefore, we do not consider citing it in our submission.

  9. The title of section 5 should be changed to "Conclusions and Future Work".

    In the revised version of the paper, we appended a separate “conclusion and outlook” section to improve the paper’s structure.

  10. The format of references should be consistent.

    We adhere to the official MDPI template at https://www.overleaf.com/latex/templates/mdpi-article-template/fcpwsspfzsph and coordinate any modifications with the editors of J.Imaging.

Round 2

Reviewer 3 Report

Comments and Suggestions for Authors

I have no further comments

Author Response

Dear reviewer, thank you your efforts in refining our initial manuscript. We believe that the revisions have significantly improved the manuscript and we appreciate your contribution in this process. 

Reviewer 4 Report

Comments and Suggestions for Authors

Thanks to the authors for incorporating the suggested revisions and improving the paper's clarity and rigor: however, I still have several concerns:

1. There are still inconsistencies in the formatting of the references. Some references have been put as endnotes, some have been put as footnotes and some are just written in the text without the entire context e.g. “Concurrently with our work, Liao et al. Published...”.

2. The authors write “we ran our method on the 6 synthetic blender scenes”. Using a limited number of synthetic scenes might raise concerns about the method's generalizability to real-world scenarios. Using only 6 scenes might not be sufficient to demonstrate the method's robustness and effectiveness. A more comprehensive evaluation would require a diverse dataset, including real-world data.

Author Response

Dear reviewer, thank you for your repeated effort in improving our initial manuscript. We took your comments into consideration and created a second revision of our paper. For detailed responses to your comments, refer to the section below:

Comment 1: There are still inconsistencies in the formatting of the references. Some references have been put as endnotes, some have been put as footnotes and some are just written in the text without the entire context e.g. “Concurrently with our work, Liao et al. Published...”.

Thank you for the clarification, we modified the revised manuscript accordingly. All footnotes and author name citations have been removed.

Comment 2: The authors write “we ran our method on the 6 synthetic blender scenes”. Using a limited number of synthetic scenes might raise concerns about the method's generalizability to real-world scenarios. Using only 6 scenes might not be sufficient to demonstrate the method's robustness and effectiveness. A more comprehensive evaluation would require a diverse dataset, including real-world data.

In our paper, we evaluate our method on (6+1) synthetic blender scenes, and on (6+5) real-world ScanNet++ scenes. The additional 5 real-world scenes were added in the first revision to better show the method's generalization capabilities. Our experiments on real-world datasets successfully showed the model's robustness and effectiveness. We consider this to be a comprehensive evaluation of our method on a diverse real-world dataset.